

# Phosphorylation/dephosphorylation response to light stimuli of *Symbiodinium* proteins: specific light-induced dephosphorylation of an HSP-like 75 kDa protein from *S. microadriaticum*

Raúl E. Castillo-Medina[1,2], Tania Islas-Flores[2] and Marco A. Villanueva[2]

[1] Posgrado en Ciencias del Mar y Limnología, Universidad Nacional Autónoma de México, Delegación Coyoacán, Ciudad Universitaria, Ciudad de México, México
[2] Unidad Académica de Sistemas Arrecifales, Instituto de Ciencias del Mar y Limnología-UNAM, Puerto Morelos, Quintana Roo, México

Corresponding author
Marco A. Villanueva,
marco@cmarl.unam.mx

## ABSTRACT

**Background.** Some genera of the family Symbiodiniaceae establish mutualistic endosymbioses with various marine invertebrates, with coral being the most important ecologically. Little is known about the biochemical communication of this association and the perception and translation of signals from the environment in the symbiont. However, specific phosphorylation/dephosphorylation processes are fundamental for the transmission of external signals to activate physiological responses. In this work, we searched phosphorylatable proteins in amino acids of Ser, Thr and Tyr from three species of the family Symbiodiniaceae, *Symbiodinium kawagutii*, *Symbiodinium* sp. Mf11 and *Symbiodinium microadriaticum*.

**Methods.** We used specific antibodies to the phosphorylated aminoacids pSer, pThr and pTyr to identify proteins harboring them in total extracts from three species of *Symbiodinium* in culture. Extractions were carried out on logarithmic phase growing cultures under a 12 h light/dark photoperiod. Various light/dark, nutritional and other stimuli were applied to the cultures prior to the extractions, and proteins were subjected to SDS-PAGE and western immunoblotting. Partial peptide sequencing was carried out by MALDI-TOF on specific protein spots separated by 2D electrophoresis.

**Results.** At 4 h of the light cycle, several Thr-phosphorylated proteins were consistently detected in the three species suggesting a genus-dependent expression; however, most Ser- and Tyr-phosphorylated proteins were species-specific. Analysis of protein extracts of *S. microadriaticum* cultures demonstrated that the level of phosphorylation of two Thr-phosphorylated proteins with molecular weights of 43 and 75 kDa, responded inversely to a light stimulus. The 43 kDa protein, originally weakly Thr-phosphorylated when the cells were previously adapted to their 12 h dark cycle, underwent an increase in Thr phosphorylation when stimulated for 30 min with light. On the other hand, the 75 kDa protein, which was significantly Thr-phosphorylated in the dark, underwent dephosphorylation in Thr after 30 min of the light stimulus. The phosphorylation response of the 43 kDa protein only occurred in *S. microadriaticum*, whereas the dephosphorylation of the 75 kDa protein occurred in the three species studied suggesting a general response. The 75 kDa protein was separated on 2D gels as two isoforms and the sequenced spots corresponded to a BiP-like protein of the

HSP70 protein family. The presence of differential phosphorylations on these proteins after a light stimulus imply important light-regulated physiological processes in these organisms.

# INTRODUCTION

The dinoflagellates of the genus *Symbiodinium* are photosynthetic microorganisms that live either freely in the seas or captured within a mutualistic symbiotic relationship in the tissues of some marine invertebrates. In both their free and symbiotic stages, they require fine sensing mechanisms to respond to changing light conditions and other stimuli from the environment that trigger signal-transduction pathways. Such signaling cascades must function with particular sets of receptor, adapter and effector proteins. One particular and fundamental switch to turn on and off signaling within the cell is through exquisitely regulated phosphorylation and dephosphorylation reactions via protein kinases and phosphatases, respectively, on key target molecules. Protein kinases are involved in all cellular functions, and in eukaryotic organisms belong to a superfamily formed by hundreds to thousands of copies. The larger protein kinase repertoire is encoded by flowering plants (600–2,500 members; over 1,000 were identified in the *Arabidopsis thaliana* genome), while a smaller number of related genes (only 518 plus 106 pseudogenes) were identified in the human genome (*The Arabidopsis Genome Initiative, 2000*; *Manning et al., 2002*; *Lehti-Shiu & Shiu, 2012*). In *Symbiodinium* however, the kinase domains are the second most abundant domains in the so far sequenced genomes, with 869 sequences (Pfam PF00069) showing a potential kinase activity (*González-Pech, Ragan & Chan, 2017*; *Liu et al., 2018*). This abundance highlights the fundamental regulatory functions of the protein kinase superfamily. For example, 177 transcripts from more than 20 serine/threonine-protein kinase families significantly changed their expression levels when *Symbiodinium* sp. (clade F, ITS2) was thermally stressed (*Giertz, Forêt & Leggat, 2017*). In another example, the general dephosphorylation of PSII proteins under stress conditions plays an important regulatory role; that is, the dephosphorylation of the D1 and D2 proteins leads to their proteolytic degradation within the PSII repair cycle in response to light stress (*Aro & Ohad, 2003*; *Baena-Gonzalez, Barbato & Aro, 1999*; *Rintamaki, Kettunen & Aro, 1996*). Additionally, a rapid dephosphorylation of the proteins of the PSII reaction center seems to be a regulatory reaction to water stress in the photosynthetic membranes of plants (*Liu et al., 2009*). On the other hand, it is believed that the dephosphorylation of LHCII promotes the association with the PSII to prevent its lateral migration within the thylakoid membrane (*Vener, 2007*), thus maintaining the LHCII *in situ* for further degradation (*Liu et al., 2009*). These reactions occur through sensing environmental signals by specific receptors at the cell membrane, transducing the signals through adapter/mediator proteins which subsequently

turn targets on and off by interaction and/or phosphorylation/dephosphorylation, to finally reach their effectors.

Photoreceptors are widely used by organisms to detect and respond to their light environment. Several putative photoreceptors, including phytochromes, cryptochromes, phototropins and rhodopsins, have been identified in the *Symbiodinium* transcript (*Xiang et al., 2015*). Cryptochromes are receptors for flavoproteins that are sensitive to blue light. They are found in plants, animals, insects, fungi and bacteria (*Chaves et al., 2011*), and have recently been shown to respond to both blue and red light in *Chlamydomonas reinhardtii* (*Beel et al., 2012*). Studies in animals and plants suggest that cryptochromes play a critical role in the generation and maintenance of circadian rhythms (*Chaves et al., 2011*). It is possible that *Symbiodinium* cryptochromes are involved in the circadian control of several light sensitive processes (*Jones & Hoegh-Guldberg, 2001*; *Sorek et al., 2014*), especially because blue is the main type of light that penetrates the water column in the oceans. Unfortunately, no detailed information on the possible light-sensing mechanisms and/or consequent signaling cascades in marine dinoflagellates is available.

Since there is very little information from the genus *Symbiodinium* regarding proteins that participate in key phosphorylation processes for signal-transduction events that arise from environmental stimuli, we sought to identify phosphorylated proteins from *S. kawagutii*, *S. microadriaticum* and *S*. Mf11. We identified a 75 kDa protein that was susceptible of changes in its phosphorylation levels upon a light stimulus. Our findings indicate that light is an important switch that impacts the phosphorylation status of several proteins in *Symbiodinium*. In particular, the 75 kDa protein may be regulated through putative conserved regulatory mechanisms of phosphorylation/dephosphorylation since its presence and light-stimulated dephosphorylation response were found in all three *Symbiodinium* species analyzed.

## MATERIAL AND METHODS

### Antibodies and reagents

Polyclonal anti-phosphothreonine (anti-pThr cat. 9381S) and anti- phosphoserine (anti-pSer cat. 2261S) antibodies were from Cell Signaling Technology[TM], Inc. (Danvers, MA). Monoclonal anti-phosphotyrosine antibodies (anti-pTyr sc-508) were from Santa Cruz Biotechnology (Dallas, TX). Monoclonal anti-actin antibody N350 was from Amersham; this antibody is commercially discontinued but we still maintain an available stock (*Villanueva, Arzápalo-Castañeda & Castillo-Medina, 2014*). Alkaline-phosphatase (AP) conjugated polyclonal anti-rabbit IgG and anti-mouse IgG antibodies raised in goat were from Zymed[TM] -Life Technologies (Grand Island, NY). The tripeptides RAD (Arg-Ala-Glu) o RGD (Arg-Gly-Glu) were from CALBIOCHEM®-Millipore (Billerica, MA, USA). Ethylene glycol-bis(2-aminoethylether)-NNN'N'-tetraacetic acid (EGTA), bovine casein hydrolyzate (N-Z-Amine A), and the amino acids, glutamic acid, arginine and glycine, were all from Sigma. Reagents 5-bromo-4-chloro-3-indolyl phosphate (BCIP) and nitro blue tetrazolium (NBT) were from Promega (Madison, WI, USA).

### Symbiodinium cell cultures

Dinoflagellate cultures of *Symbiodinium kawagutii* (referred as *S. kawagutii*) were a kind gift of Dr. Robert K. Trench (University of California at Santa Barbara). *Symbiodinium* sp. Mf11.5b.1 (referred as *S.* Mf11), and *Symbiodinium microadriaticum* Subsp. *microadriaticum* (also known as MAC-CassKB8 and from now on referred to as *S.* KB8) originally isolated from the jellyfish *Cassiopea xamachana,* were a kind gift of Dr. Mary Alice Coffroth (State University of New York at Buffalo). These cell lines correspond to clades F, B and A, respectively; they were routinely maintained in our laboratory in ASP-8A medium under photoperiod cycles of 12 h light/dark at 25 °C. Light intensity was maintained at 80–120 $\mu$mole quanta m$^{-2}$ s$^{-1}$.

### Preparation of total protein extracts for phospho protein screening and responses to stimuli

Six-d-old *Symbiodinium* cells in ASP-8A culture medium were collected 4 h after the light phase of the photoperiod initiated, and were concentrated by centrifugation at $1,400 \times g$ for 8 min. The cell pellet was suspended with 500 $\mu$l of 1× Laemmli buffer (60 mM Tris-HCl, pH 6.8, 1% (w/v) sodium dodecyl sulfate (SDS), 10% (v/v) glycerol, 3% (v/v) β-mercaptoethanol, 0.02% bromophenol blue) (*Laemmli, 1970*), supplemented with 0.2 mM NaVO$_3$, 10 mM NaPPi and a cocktail of protease inhibitors (Complete®; Roche, Basel, Switzerland), and placed in screw-cap polypropylene vials with a 0.25 ml total volume of glass beads (465–600 $\mu$m diameter) previously cooled. Subsequently, the cells were lysed with vigorous shaking in a MINI-BEAD BEATER$^{TM}$ (Biospec products) at maximum speed for 3 min. Then, the cell lysate was heated at 95 °C for 5 min, and centrifuged at $16,000 \times g$ for 10 min. The supernatant was used for analysis in polyacrylamide gels by SDS-PAGE (sodium dodecyl sulfate-polyacrylamide gel electrophoresis) and western blot.

### Protein electrophoresis in SDS-PAGE gels and western blot

The protein extracts were separated in discontinuous denaturing gels (*Laemmli, 1970*), of 10% polyacrylamide in the separation zone [375 mM Tris-HCl, pH 8.8; 10% acrylamide/bis-acrylamide; 0.1% SDS; 0.1% ammonium persulfate (APS); 0.106% N, N, N, N'-tetramethylethylenediamine (TEMED)], and 4% polyacrylamide in the stacking zone [125 mM Tris-HCl, pH 6.8; 4% acrylamide; 0.137% bisacrylamide; 0.1% SDS; 0.1% APS; 0.066% TEMED], in a Mini-PROTEAN®3 System (Bio-Rad, Hercules, CA, USA). After electrophoresis, the proteins were transferred to PVDF membranes in "friendly buffer" (25 mM Tris-HCl, 192 mM glycine, 10% isopropanol, *Villanueva, 2008*) at a constant current of 300 mA for 1 h. The membranes were blocked in a solution of 3% bovine serum albumin (BSA) in PBS (2.79 mM NaH$_2$PO$_4$, 7.197 mM Na$_2$HPO$_4$, 136.9 mM NaCl, pH 7.5) for 1 h at 50 °C, with gentle agitation. After blocking, the primary antibodies, anti-pThr (1:2,500), anti-pSer (1:500), anti-pTyr (1:500) or anti-actin (1:1,000) diluted in PBS containing 0.01% Triton X-100 (PBS-T), were added to the membranes and incubated overnight with gentle rocking at room temperature. Then, the membranes were washed five times, 5 min each, in PBS-T, and incubated with the appropriate secondary antibodies (alkaline-phosphatase conjugated anti-rabbit IgG or anti-mouse

IgG) at 1:2,500 dilution for 2 h at room temperature. Subsequently, the membranes were washed again five times, 5 min with PBS-T. Finally, they were developed with a commercial solution of 5-bromo-4-chloro-3-indolyl phosphate (BCIP) and nitro blue tetrazolium (NBT) according to the manufacturer (Promega, Madison, WI, USA) in PBS-T for the primary polyclonal antibodies to ensure more astringency and less background. In the case of primary monoclonal antibodies, the final wash was followed by a brief rinse with alkaline developing solution (100 mM Tris-HCl, 150 mM NaCl, 1 mM $MgCl_2$, pH 9), and development in the same buffer. Western blot incubations, washing and development were also carried out in PBS-T or TBS-T (20 mM Tris, 150 mM NaCl, 0.01% TX-100, pH 7.6) with identical results.

## Two-dimensional gel electrophoresis and partial peptide sequencing

Six-d-old cell cultures of *S*. KB8 collected 2 h before the dark phase of the photoperiod, were concentrated by centrifugation and suspended in 50 ml of fresh ASP-8A medium. After a 12 h incubation in the dark, the cells were precipitated by centrifugation at $2,600 \times g$, 5 min, 26 °C and suspended in 10 ml of 0.2 mM $NaVO_3$ and 10 mM NaPPi in PBS-T added with a Complete® (Roche, Basel, Switzerland) cocktail of protease inhibitors, taking special care to carry out the whole procedure under dark conditions. The cells were then disrupted with a French press (SLM-AMINCO French® pressure cell press) at a pressure of 20,000 psi. Then, the cell lysate was chilled on ice and centrifuged at $100,000 \times g$, 1 h to recover the proteins from the supernatant.

Next, the protein pellet was dissolved with seven volumes of PB (2.79 mM $NaH_2PO_4$, 7.197 mM $Na_2HPO_4$, pH 7.6) and applied to a column with 1 ml matrix volume of DEAE-sephacel previously equilibrated with 50 ml 0.02 M NaCl in PB. Then, the column was washed with 10 ml of the same buffer and eluted with 2 ml of 0.4 M NaCl in PB. The eluted proteins were precipitated with 8 volumes of cold acetone (at −20 °C), and one volume of trichloroacetic acid (TCA) for 1 h at −20 °C. Then, the mixture was centrifuged at $5,000 \times g$ for 5 min and the protein pellet was washed twice with 2 ml of acetone at −20 °C. Afterwards, the proteins were dissolved in BioRad® ReadyPrep buffer (8 M urea, 2% CHAPS, 50 mM DTT, 0.2% w/v ampholytes 3/10 BioLite® from BioRad and bromophenol blue), and the solution was used to hydrate a 7 cm IPG strip (ReadyStrip pH 4-7; Bio-Rad, Hercules, CA, USA) for 12 h at 18 °C. Then, the proteins adsorbed in the strip were separated with a programmed current of 250 V, 20 min, 4,000 V, 2 h and 4,000 V to cover 10,000 V-h in a PROTEAN IEF cell (Bio-Rad, Hercules, CA). Subsequently, the proteins in the strips were first denatured in 1.5 ml of Buffer I® (Bio-Rad; 6 M urea, 2% SDS, 0.375 M Tris-HCl pH 8.8, 20% glycerol and 2% w/v DTT) for 15 min, and then in 1.5 ml of Buffer II® (Bio-Rad; 6 M urea, 2% SDS, 0.375 M Tris-HCl pH 8.8, 20% glycerol added with 2.5% w/v iodoacetamide) for 15 min. Then, the strip was briefly rinsed in running buffer (25 mM Tris, 192 mM Glycine, 0.1% SDS, pH 8.3.) and loaded onto a 10% polyacrylamide SDS-PAGE gel to separate the proteins in the second dimension. Finally, the proteins were either stained with colloidal coomassie blue, or transferred onto a PVDF membrane for analysis with anti-pThr antibodies. The gel spots corresponding to the anti-pThr immunodetected signal were excised with a pipet tip, placed in a sterile

eppendorf tube with 50 µL milliQ water, and sent for partial peptide sequencing to the Unit of Proteomics of the Institute of Biotechnology from UNAM in Cuernavaca, Morelos, México. Each spot was also reanalyzed by western blot with the anti-pThr antibodies to ensure their identity. In parallel, an adjacent mock spot corresponding to a major Mr ∼77 kDa protein was included as negative control for the analysis. The most probable sites of phosphorylation in threonine from the sequences were predicted by the NetPhosK server (*Blom et al., 2004*).

## Light treatments on *Symbiodinium* cells adapted to darkness

Six-d-old cultures from *S.* KB8, *S.* Mf11 and *S. kawagutii* from three biological replicates collected 2 h before the dark phase of the photoperiod, were concentrated and suspended in 40 ml of fresh ASP-8A medium to reach 0.8–1.4 $\times 10^6$ cells/ml. Equal aliquots of each phylotype were placed in four 15 ml Falcon tubes wrapped with aluminum foil. The cells in the tubes were allowed to adapt in the dark (12 h) during their night cycle. After this period, the cells from one of the tubes were concentrated and broken with glass beads without exposing them to light throughout the process (control), and the other three tubes were released from the aluminum foil to expose them to light. Protein extractions with glass beads were performed at 30, 60 and 240 min after light exposure, and the supernatant from the lysate was heated at 95 °C, 5 min, and used for the analysis. Equal loads of proteins were adjusted with equal aliquots of cells and also standardized by staining with coomassie blue in 10% polyacrylamide SDS-PAGE gels. Finally, the extracts were analyzed by western blot which included actin as an additional internal loading control. The bands from the triplicate samples were captured with a ChemiDoc-It2 Imager (UVP-Analytik Jena, Upland, CA, USA) and analyzed by densitometry with the system's VisionWorks LS software, normalized with the internal actin control to obtain an arbitrary level of intensity for each antibody reaction. The results were integrated into graphs displaying the average band intensity of the three biological replicates for each treatment.

## Treatments of *Symbiodinium* cells with various stimulating agents

Six-d-old *S.* KB8 cells from a 1 L culture were collected by centrifugation at $1,400 \times g$ for 8 min, and suspended in 10 ml fresh medium, they were then divided into 9 aliquots of 1 ml in 15 ml Falcon tubes. The cells were concentrated again by centrifugation as above and the pellets were resuspended in: (a) 10 ml of fresh ASP-8A medium (control); or (b) 10 ml fresh medium supplemented with either 0.1% (w/v) casein, 10 mM glycine, 10 mM arginine, 10 mM glutamic acid, 100 µg/ml of the tripeptides RAD (Arg-Ala-Glu), RGD (Arg-Gly-Glu) (CALBIOCHEM®; Millipore, Billerica, MA, USA), 15 mM EGTA (ethylene glycol-bis (2-aminoentylether)-NNN'N'-tetra acetic acid), or 20 mM CaCl$_2$. All samples were incubated for 30 min at room temperature and after the incubations, the cells were concentrated by centrifugation followed by protein extraction with glass beads using the Bead Beater. Equal loads of proteins were analyzed by protein staining on SDS-PAGE and western blot with anti-pThr antibodies.

## RESULTS

### Anti-pTyr, -pSer and -pThr antibodies immunodetect several phosphoproteins from *S. kawagutii*, *S.* KB8 and *S.* Mf11 total extracts

Protein extracts from the three phylotypes under study were analyzed with the three anti-phosphoaminoacid antibodies by western blot. Anti-pTyr antibodies revealed intense bands with Mr ~32 kDa in all three species (asterisk in Fig. 1C, lanes 1–3, respectively, and Fig. S1C), whereas species-specific bands of Mr's ~43, 57, 62 and 77 kDa were observed for *S.* KB8 (Fig. 1C, lane 1, and Fig. S1C, lane 1); of Mr's ~30 and 54 kDa for *S.* Mf11 (Fig. 1C, lane 2, and Fig. S1C, lane 2); and of Mr's ~48, 51 and 79 kDa for *S. kawagutii* (Fig. 1C, lane 3, and Fig. S1C, lane 3), indicating that these bands corresponded to proteins phosphorylated on Tyr residues but were not ubiquitous, except for the Mr ~32 kDa protein. Anti-pSer antibodies revealed intense bands with Mr ~65 kDa in all three species (Fig. 1B, asterisk in lanes 1–3, and Fig. S1B, lanes 1–3), and a weaker Mr ~35 kDa band also in the three species analyzed (asterisk in Fig. 1B, lanes 1–3, and Fig. S1B, lanes 1–3). A relatively high number of weaker bands both ubiquitous and species-specific, were also detected with these antibodies in the three species analyzed (Fig. 1B, lanes 1–3, and Fig. S1B, lanes 1–3). These results indicated that a considerable number of proteins from all three species were phosphorylated on Ser residues. Finally, anti-pThr antibodies revealed several phospho proteins that were present in all three species analyzed. Here, we observed weak bands of Mr's ~43, 46, 50 and 55 kDa (asterisks in Fig. 1C, lanes 1, 2 and 3), and two clear bands with Mr's ~75 (double asterisk in Fig. 1A, lanes 1–3, and Fig. S1A, lanes 1–3) and 91 kDa (asterisk in Fig. 1A, lanes 1–3, and Fig. S1A, lanes 1–3). Additionally, the anti-pThr antibodies detected a clear band of Mr ~29 kDa present in *S.* KB8 and *S.* Mf11 (Fig. 1A, lanes 1 and 2, and Fig. S1A, lanes 1 and 2) as well as a clear Mr ~31 kDa protein band present only in *S. kawagutii* (Fig. 1A, lane 3, and Fig. S1A, lane 3), and a Mr ~107 kDa protein band present only in *S.* KB8 (Fig. 1A, lane 1, and Fig. S1A, lane 1). All the proteins found to cross-react with the anti-phospho amino acid antibodies and their corresponding Mr's are shown in Fig. S1. These data indicated: (a) the presence of proteins also phosphorylated on Thr residues; and (b) that there were phosphorylated proteins on the amino acids Tyr, Ser and Thr, in all three *Symbiodinium* species. One protein of Mr ~75 kDa, which at this point we named pp75, fulfilled our criteria of being present in all three species and presenting a clear reproducible reaction upon developing by western blot. Therefore, we chose to further analyze this protein for possible changes upon treatment of the cells with light/dark and other stimuli.

### Some phosphorylated proteins in threonine responded to a light stimulus

Analysis by western blot with anti-pThr in cells from all three species adapted to darkness during the dark phase of the 12-h photoperiod showed that pp75 maintained a considerable degree of apparent phosphorylation under this condition (Figs. 2A–2C, lanes 1; arrow labeled 75). Conversely, when the cells were stimulated with light for 30 min and analyzed by western blot, the intensity of pp75 decreased significantly (Figs. 2A–2C, lanes 2; arrow labeled 75). The weak intensity of the band was maintained even after 60 and 240 min

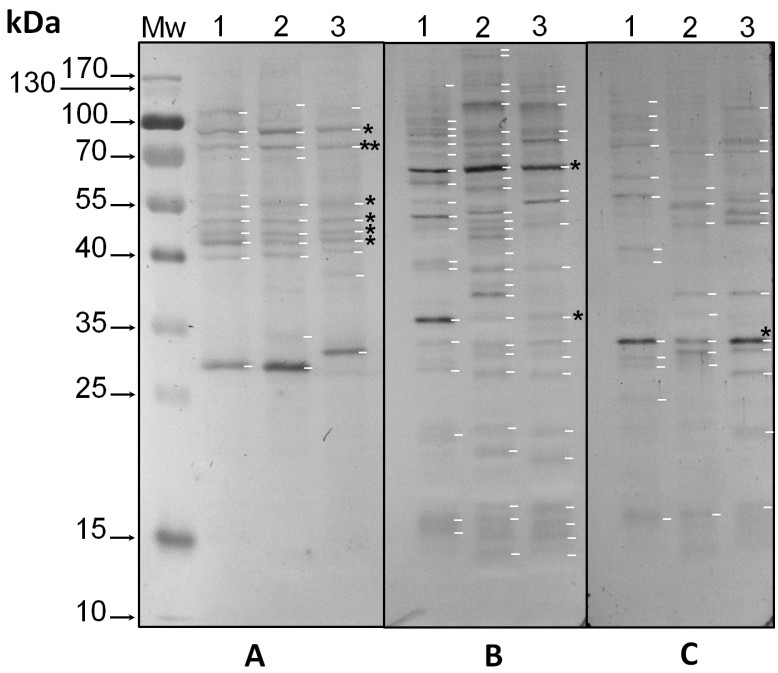

**Figure 1** **Immunodetection by western blot of phospho amino acids present in proteins of three species of *Symbiodinium* with anti-pThr (A), anti-pSer (B) and anti-pTyr (C) antibodies.** In the figure, the lanes correspond to: molecular weight markers (Mw); protein extracts of *Symbiodinium* KB8 (lanes 1), *Symbiodinium* Mf11 (lanes 2) and *Symbiodinium kawagutii* (lanes 3). The asterisks within the figure indicate protein bands immunodetected in all three species and the arrows to the left indicate the relative molecular weights in kDa. pp75 is denoted with a double asterisk in A.

of exposure to light (Figs. 2A–2C, lanes 3 and 4, respectively; arrow labeled 75). These results indicated that this protein, during the dark phase, is significantly phosphorylated on Thr; however, it dephosphorylates after 30 min of a ''shock'' of light and remains dephosphorylated for at least 240 min of such light exposure. Interestingly, the opposite behavior was clearly observed for a Mr ∼43 kDa protein only in *S*. KB8 cells before the light stimulus. This protein was not observed as phosphorylated in Thr when the cells were adapted to darkness (Fig. 2A, lane 1; arrow labeled 43). However, after 30 min of light exposure, the anti-pThr antibodies detected an intense band, indicating an enhanced phosphorylated state (Fig. 2A, lane 2; arrow labeled 43), which was maintained even after 60 and 240 min of exposure to light (Fig. 2A, lanes 3 and 4, respectively; arrow labeled 43). Species-specific light-stimulated phosphorylation behaviors were observed for a number of proteins immunodetected with anti-pThr antibodies. One Mr ∼29 kDa protein with similar behavior was detected in *S*. KB8 and *S*. Mf11 (Fig. 2A and B, lanes 1–4; arrow labeled 29), two of Mr's ∼40 and 107 kDa in *S*. KB8 (Fig. 2A, lanes 1–4; arrows labeled 40 and 107, respectively); and finally, one of Mr ∼31 kDa from *S. kawagutii* (Fig. 2C, lanes 1–4; arrow labeled 31). The dephosphorylation of pp75 on Thr was consistently and reproducibly observed in cells from all three species when the light stimulus was applied in comparison to the cells adapted to darkness before the stimulus. The revealed bands
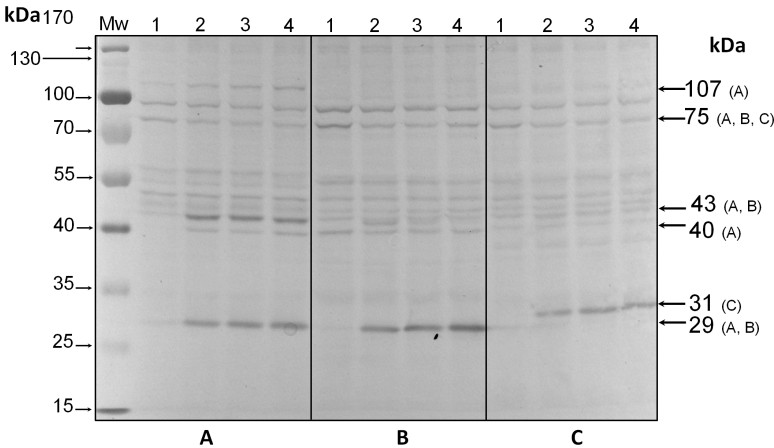

**Figure 2** Immunodetection by western blot with anti-pThr antibodies of pp75 from *in vitro* cultures of *Symbiodinium* KB8 (A), *Symbiodinium* Mf11 (B) and *Symbiodinium kawagutii* (C), incubated for 12 h in darkness (lanes 1) and after stimulation with light for 30 (lanes 2), 60 (lanes 3) and 240 (lanes 4) min.

were specific and the whole membrane showed specific reactions of the 75 and 43 kDa bands upon development of the blot (Fig. 2). In both cases, we quantified the response by densitometric analysis of the blots using triplicate biological samples and the intensity of the bands was normalized to the reaction of an anti-actin monoclonal antibody as internal control. The results were normalized, averaged, subjected to ANOVA, and graphed. In all cases, a statistically significant decrease in the phosphorylation level of pp75 in at least one time point after the light treatment was detected for all three species analyzed (Fig. 3A; S. KB8, lane 3; S. Mf11, lanes 2 and 3; S. kw, lane 4). The same analysis was carried out on the 43 kDa protein with similar results showing the opposite behavior for S. KB8 but not for the other two species (Fig. 3B). Although we cannot assure that they are the same as the 43 kDa protein that responds to light in S. KB8, Mr ~43 kDa proteins from S. Mf11 and S. kawagutii cells, also recognized by the anti-pThr antibodies did not change their level of apparent phosphorylation after equivalent times after the light stimulus (Figs. 2B and 2C, lanes 2–4; Fig. 3B, S. Mf11 and S. kw, respectively), compared to the control (cells adapted to darkness, Fig. 2B and C, lanes 1; Fig. 3B, S. Mf11 and S. kw, respectively). This indicated that the Mr ~43 kDa protein may have a species-specific behavior and only pp75 fulfilled the criterion of ubiquity in at least all three *Symbiodinium* species analyzed.

## Threonine-phosphorylated proteins from *S*. KB8 did not show differential phosphorylation in response to other stimuli

In addition to light, we sought to stimulate S. KB8 cells in culture in order to observe possible changes in phosphorylation levels of proteins and relate those changes with other metabolic or signal-transduction pathways. Nevertheless, no changes were detected upon any of the treatments. None of the nutritional stimuli provided by incubation with the amino acids: glycine, arginine, glutamate, or a casein hydrolyzate caused significant changes in the degree of apparent phosphorylation in Thr of any protein including pp75 (Fig. S3,

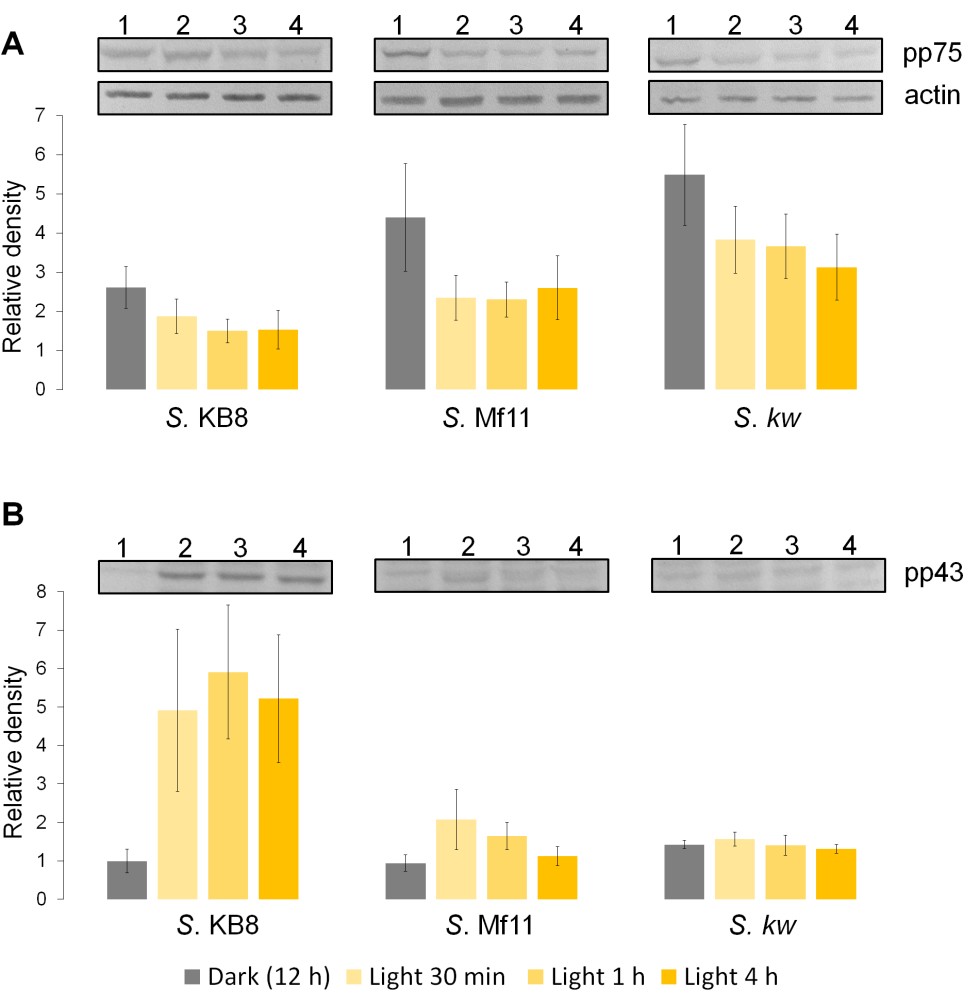

**Figure 3 Changes in the phosphorylation level of three independent biological replicates determined by western blot and densitometric analysis with anti-pThr and anti-actin monoclonal antibodies (panel labeled "actin" shown as example) used as internal loading control for normalization.** The figure shows the bands corresponding to pp75 (A), and the 43 kDa (B) protein from *in vitro* cultures of *Symbiodinium* KB8 (*S.* KB8), *Symbiodinium* Mf11 (*S.* Mf11) and *Symbiodinium kawagutii* (*S. kw*), after the cells were incubated for 12 h in darkness (lanes 1) and analyzed after light stimulation for 30 (lane 2), 60 (lane 3) and 240 (lane 4) min. The bars in the graphs represent the average of the relative density for each band normalized with the density of the actin band (panel labeled "actin").

lanes 2, 3, 4 and 5, respectively), compared to the control (Fig. S3, lane 1). In parallel, extracellular matrix stimuli were tested with the tripeptide RGD (Arg-Gly-Asp), which is an inhibitor of the adhesion of integrins to their receptors. These RGD tripeptides did not cause any change on the phosphorylation of the analyzed proteins in *S.* KB8 cells (Fig. S3, lane 6), compared to the control without stimulation (Fig. S3, lane 1). As expected, the non-active tripeptide control RAD (Arg-Ala-Asp), did not cause any significant effect either (Fig. S3, lane 7). Finally, no alterations were observed in the phosphorylation pattern when the $CaCl_2$ concentration was increased in the medium (Fig. S3, lane 8), or when calcium ions were sequestered with EGTA (Fig. S3, lane 9), compared to the control (Fig. S3, lane 1).

These results support the role of specific pThr phosphorylation as part of a light-regulated response in *Symbiodinium*.

### The threonine-phosphorylated 75 kDa protein from *S.* KB8 is a BiP-like protein of the HSP70 protein family

The light-responding pp75 from *S.* KB8 was enriched through anion exchange column chromatography on DEAE-sephacel and the eluted fraction precipitated with ice-cold acetone-TCA, followed by analysis and separation by two-dimensional gel electrophoresis as described in 'Materials and Methods'. The western blot of the separated proteins immunodetected with anti-PThr antibodies identified two Mr ∼75 kDa phosho protein spots of pI's ∼5.4 and 5.6 (Fig. 4A, spots 1 and 2, respectively), indicating the presence of at least two isoforms of pp75. Only a few spots corresponding to the Mr's of the phospho proteins previously immunodetected by the anti-PThr antibodies from *S.* KB8 extracts (Fig. S1A, lane 1) were observed, whereas a complex mixture of proteins was observed in the equivalent coomassie blue-stained 2D gel (Fig. 4B, spots 1 and 2). In addition, the excised spots as well as an adjacent mock spot (Fig. 4B, M) corresponding to a major protein of similar Mr, were re-analyzed by western blot with anti-PThr antibodies and only the lanes corresponding to the phospho protein spots were immunostained (Fig. S4, lanes 1 and 2), while the mock spot showed no cross-reaction (Fig. S4, lane 3). Both spots were sent for a partial peptide sequencing service and the identity from either spot matched a BiP-like protein sequence belonging to the HSP70 protein family from the *S. microadriaticum* transcriptome with 100% identity (Fig. 4C, Spot 1 and 2 sequences in red; Accession No. OLP91134). It is important to note that the reported sequence displayed a putative molecular weight of >200 kDa for the translated product, while pp75 migrates at Mr ∼75 kDa. Therefore, we analyzed this sequence further by comparing the corresponding genomic and transcriptomic sequences. This comparison revealed the correct start ATG and stop codons that resulted in the correctly translated product of 75 kDa. This corrected sequence is shown in Fig. 4C and it was used for a BLAST analysis against the *S. microadriaticum* transcriptome, which yielded at least other four related sequences encoding HSP70-like proteins (with long enough sequence showing relevant domains; accession numbers OLQ05431, OLP79133, OLP80221 and OLP86850), and one likely related short sequence (Accession No. OLP81269). Three of these sequences, including pp75, corresponded to BiP-like proteins (Accession No's. OLP91134, OLP86850 and OLP81269). The sequence also contained 6 potential phosphorylated threonine residues (Fig. 4C, green Thr residues). These results indicated that the Mr ∼75 kDa protein from *S.* KB8 displaying differential phosphorylation on Thr upon a light stimulus was inequivocally identified as a BiP- or HSP70-like protein. Therefore, we renamed the protein as SmicHSP75.

## DISCUSSION

*Symbiodinium* usually lives as endosymbiont in reef hosts (*Muscatine & Porter, 1977*). Due to its photosynthetic nature, it must rely on fine sensing mechanisms to respond to changing light conditions in its environment, which must also be transduced to molecular signals that become physiological responses. Phosphorylation and dephosphorylation events are major

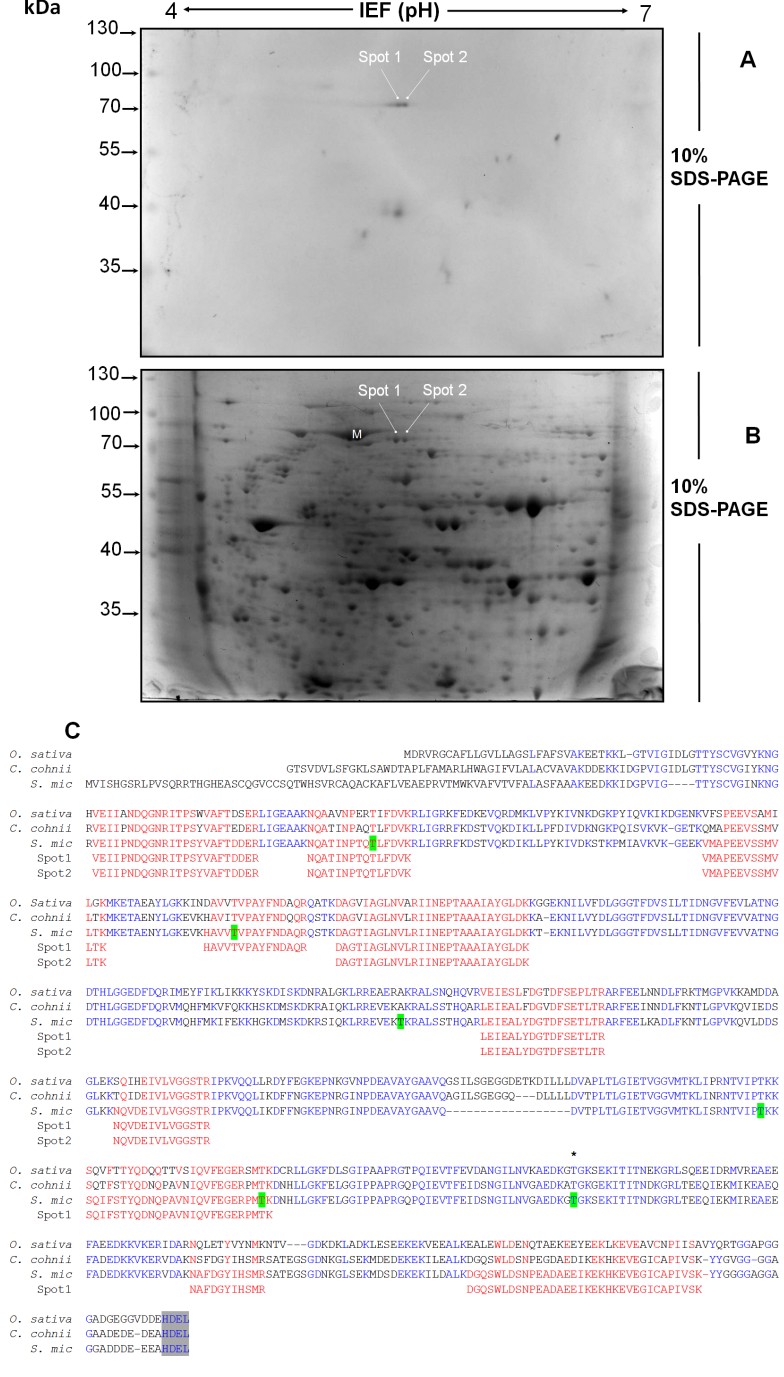

**Figure 4** **Separation by two-dimensional (2D) gel electrophoresis (IEF + SDS-PAGE) and immunodetection of SmicHSP75 from *S*. KB8 enriched protein extracts for subsequent isolation and partial peptide sequencing.** (A) Western blot of the resolved proteins on the 2D gel of DEAE-sephacel enriched extracts incubated after 12 h darkness. The two spots labeled spot 1 and spot 2 correspond to two SmicHSP75 isoforms of pI's 5.4 and 5.6, respectively. (B) An equivalent 2D gel stained with coomassie blue where the SmicHSP75 isoforms are located (spot 1 and 2, respectively). 

**Figure 4 (…continued)**
(C) Multiple alignment of the partial sequences of the peptides obtained from both spots with the
HSP70/BiP sequences of *Oryza sativa* (accession number Q6Z7B0), *Cryptecodinium cohnii* (accession
number Q8S4R0), and *Symbiodinium microadriaticum* (*S. mic*; accession number OLP91134). The
figure shows the motif of retention in the endoplasmic reticulum (*Munro & Pelham, 1987*) highlighted
in gray and characteristic of the BiP; highlited in green are the most probable sites of phosphorylation
in threonine predicted by the NetPhosK server (*Blom et al., 2004*); in addition, the asterisk indicates
a conserved phosphorylatable threonine presumably involved in protein synthesis regulation in
*Chlamydomonas reinhardtii* (*Díaz-Troya et al., 2011*).

cellular switches that trigger cellular responses upon environmental stimuli and are likely
to play an important role in such fine sensing mechanisms (*Graves & Krebs, 1999*). Since
there is very little information regarding proteins that participate in key phosphorylation
processes for signal-transduction events from the genus *Symbiodinium*, we sought to
identify phosphorylated proteins on the amino acids, threonine, serine and tyrosine from
three different *Symbiodinium* species, *S. kawagutii*, *S.* KB8 and *S.* Mf11. We further sought to
identify a protein that was susceptible of changes in its phosphorylation levels upon various
types of stimulus. Several proteins containing phosphorylated amino acids were identified,
although there were variations in the three species analyzed. Therefore, we chose to focus
our analysis on proteins that fulfilled two criteria. First, that their cross-reaction with the
antibody was clear and reproducible. Second, that their presence was not species-specific
so that they were observed in all three species analyzed. Our first screening with anti-pThr,
anti-pSer and anti-pTyr antibodies, revealed several candidate proteins (Fig. S1), indicating
that phosphorylation is an important feature of *Symbiodinium* proteins, which is also
consistent with the hypothesis that such proteins could be key players of signaling events.
A first attempt revealed only a few bands; however, we found two critical conditions that
improved the signal-to-noise ratio significantly: first, when we used PBS-T for development
of the incubations with polyclonal primary antibodies but alkaline developing solution
for monoclonal primary antibodies (see 'Materials and Methods'), more and stronger
bands appeared; second, the use of PVDF instead of nitrocellulose as transfer media
increased the band resolution significantly. Therefore, we used these conditions for all
subsequent western blot experiments. The proteins detected with anti-pThr antibodies
with Mrs ∼43, 46, 50, 55, 75 and 91 kDa, were present in all three species (Fig. 1A and
Fig. S1A). A relatively high number of bands both ubiquitous and species-specific were
also detected with anti-pSer antibodies (Fig. 1B and Fig. S1B), as well as with anti-pTyr
antibodies (Fig. 1C and Fig. S1C), but none fulfilled the criteria of both clear identification
and presence in all three species. Conversely, one protein cross-reactive with anti-pThr
antibodies, that of Mr ∼75 kDa (pp75), that was present in all three species, gave a strong
cross-reaction with the anti-pThr antibodies (Fig. 1A), and underwent significant changes
in its phosphorylated state upon a light stimulus (Figs. 2A and 3A). This protein displayed at
least two isoforms of the same molecular weight (Fig. 4A, spot 1 and 2), as immunodetected
with the anti-pThr antibodies likely indicating a differential degree of phosphorylation.
This is consistent with the modifications in the phosphorylation level detected upon light
exposure of the cells (Fig. 3A). The partial peptide sequencing of this protein allowed its

inequivocal identification as a BiP-like protein of the HSP70 protein family (Accession No. OLP91134) from *S. microadriaticum*. We therefore re-named pp75 as SmicHSP75. Since the annotated transcriptomic sequences are uncured raw sequences, and the one corresponding to SmicHSP75 yielded a translated product of >200 kDa, we took the full SmicHSP75 genomic sequence from the *S. microadriaticum* genome (*Aranda et al., 2016*) and used it to obtain the correct amino acid sequence (Fig. 4C). When we used this corrected sequence for BLAST analysis against the transcriptome, we detected at least six isoforms of the protein, as well as a conserved Thr in such sequences (Fig. 4C, asterisk). This conserved Thr has been reported as a target of phosphorylation in *C. reihardtii*, involved in metabolic regulation upon a nutritional stimulus (*Díaz-Troya et al., 2011*). In addition, we detected six potential phosphorylated threonine residues (Fig. 4C, green Thr residues), which are consistent with the possibility that the identified isoforms could correspond to different combinations of phosphorylations. The subsequent finding that this protein was originally phosphorylated on Thr during the dark phase of the *Symbiodinium* growth cycle, and was dephosphorylated upon light exposure, indicated that light was an important signal that triggered a number of signaling cascades that lead to its dephosphorylation. This also implied the existence of a phosphatase that must be able to rapidly respond to the light stimulus and act upon SmicHSP75. Another protein of Mr ~43 kDa present in all three species and cross-reactive with anti-pThr antibodies appeared to undergo a gradual enhanced phosphorylation upon light exposure but this modulatory effect was detected exclusively in *S.* KB8 (Fig. 3B, *S.* KB8). This suggested that it was part of a species-specific process; therefore, our subsequent analyses were performed exclusively on SmicHSP75, which was present in all three species.

We do not know which other proteins participate both upstream and downstream of the signaling cascade of SmicHSP75 but the dephosphorylation response also implies the existence of a photoreceptor which transmits the signal to other relay molecules that lead to the action of the yet unknown phosphatase. Similarly, unknown molecules downstream of SmicHSP75 must exist, to transduce the signal into a physiological response. The fact that the protein is highly phosphorylated during the dark phase and a dephosphorylation occurs upon the light stimulus, points towards two other important implications: first, there must be a kinase that phosphorylates the protein at the end of the light phase and during the dark phase of the cycle; and second, there must also be a phosphatase that responds upon the activation of the molecular switch that is triggered after 30 min of a light stimulus. This is also suggestive of a mechanism that puts the cell into a new metabolic (photosynthetic) state after the light is turned on. This may also reflect a light-sensing mechanism occurring under the natural state of the free-living and/or endosymbiotic cells in the water column after sunrise. It has been reported that blue light is most likely the type of radiation that enters the water column (*Braun & Smirnov, 1993*) and perception mechanisms by chryptochromes are likely to be at play under those conditions. We are currently searching for the companion proteins of SmicHSP75 in this interesting light-modulated interplay.

Besides light, other stimuli might also cause changes in phosphorylation on the same protein. We tested a number of such stimuli that included nutritional (casein hydrolyzate and amino acids), extracellular matrix stimulating peptide (RGD), and calcium exposure

and depletion ($Ca^{2+}$ and EGTA). In none of the cases significant dephosphorylation of SmicHSP75 or enhanced phosphorylation of the 43 kDa protein were observed (Fig. S3). This indicated that *Symbiodinium* cells are much more sensitive to light than other stimuli, which is also consistent with their photosynthetic nature.

Our results represent pioneer studies on proteins that are likely to participate in light modulated signal-transduction events and that have not been studied in these symbiotic photosynthetic microorganisms to date. Future research will focus on the detailed functional characterization of this light-responsive protein target and its companion players in the light-sensing mechanism.

## CONCLUSIONS

We identified a 75 kDa protein (SmicHSP75) cross-reactive with anti-pThr antibodies, that was present in the three species of *Symbiodinium*, *S.* KB8 (*S. microadriaticum*), *S. kawagutii* and *S.* Mf11. The protein was significantly phosphorylated at the end of the dark phase of the growth cycle but underwent a significant decrease in its phosphorylated state after 30 min of a light stimulus, and remained hypophosphorylated even after 260 min. This phosphorylation/dephosphorylation behavior indicated that light was an important signal that triggered a number of signaling cascades that are likely to be transient and this protein could be part of a mechanism that warns the cell to switch to a new (photosynthetic) metabolic state. The protein was inequivocally identified as a BiP-like protein of the HSP70 protein family.

## ACKNOWLEDGEMENTS

We are grateful for the technical help of MI Miguel Ángel Gómez-Reali.

### Funding

This work was supported by grants PAPIIT IN210514 and PAPIIT IN203718 from the National Autonomous University of México (DGAPA-UNAM), and 285802 from the National Council of Science and Technology (CONACyT) to Marco A. Villanueva. Raúl E. Castillo-Medina was supported by fellowship 255464 from CONACyT. The Association of Marine Laboratories of the Caribbean paid the open access fee for publication of this work. The funders had no role in study design, data collection and analysis, decision to publish, or preparation of the manuscript.

### Grant Disclosures

The following grant information was disclosed by the authors:
National Autonomous University of México (DGAPA-UNAM): PAPIIT IN210514, PAPIIT IN203718.
National Council of Science and Technology (CONACyT): 285802.
CONACyT: 255464.

## Competing Interests

The authors declare there are no competing interests.

## Author Contributions

- Raúl E. Castillo-Medina and Tania Islas-Flores conceived and designed the experiments, performed the experiments, analyzed the data, prepared figures and/or tables, authored or reviewed drafts of the paper, approved the final draft.
- Marco A. Villanueva conceived and designed the experiments, analyzed the data, contributed reagents/materials/analysis tools, prepared figures and/or tables, authored or reviewed drafts of the paper, approved the final draft.

## Data Availability

The raw measurements of the densitometric determinations are available in File S5. In addition, the figures shown in the article were assembled with the original, non-manipulated western blots.

## Supplemental Information

Supplemental information for this article can be found online at http://dx.doi.org/10.7717/peerj.7406#supplemental-information.

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
