# Peer review of "Phosphorylation/dephosphorylation response to light stimuli of Symbiodinium proteins: specific light-induced dephosphorylation of an HSP-like 75 kDa protein from S. microadriaticum"

_PeerJ, doi:10.7717/peerj.7406_

## Round 0.1 · original submission · Major Revisions

Please carefully address all the critiques of all reviewers and revise your manuscript accordingly.

·

Basic reporting

I feel the manuscript lacks a bit of general introduction or discussion about the current knowledge on the roles and evolution of kinases in the plant kingdom. In particular, the evolutionary conservation of protein kinases in eukaryotes is not mentioned at all. For example, a quick search in PFAM database retrieved more than 600 Symbiodinium protein sequences with a potential kinase activity. Also, protein kinases are well described in multicellular plants (for example Arabidopsis thaliana has >1000 genes encoding kinases, see Wang et al., “The Protein Phosphatases and Protein Kinases of Arabidopsis thaliana”, Arabidopsis book, 2007).

I also found some typos in the manuscript:
- line 60 “… genus Symbiodinium, are photosynthetic…" I would remove the coma;
- idem line 62 “In both, their…”; line 234: “(gives)”;
- lines 302-303, I did not understand the following sentence “Unfortunately, because anti-pTyr antibody-cross reactive proteins were inconsistently observed and due to the lack of reproducibility, were not considered."

Experimental design

Major technical concern: No loading controls are provided in the figures. I strongly encourage the authors to provide images of nitrocellulose membranes stained with a general protein staining such as coomassie blue to show that equal amounts of proteins are present in the lanes analyzed by western blot. This is of particular importance for the experiments where modulation of the phosphorylations is induced experimentally. The presence of such loading controls would greatly improve the reader’s level of confidence in the experiments.

Minor technical concern: the authors are using a NBT/BCIP revelation technique for the revelation step of the western blots. This technique is based on the activity of an alkaline phosphatase and I am questioning the effect of this phosphatase on nitrocellulose immobilized phosphorylated proteins.

Validity of the findings

In this study, Castillo-Medina and colleagues have used an interesting approach to reveal the presence of serine-, threonine- and proline-phosphorylated proteins in extracts from different Symbiodinium species subjected to various experimental conditions. A strong limitation of the technique lies in the fact that there is no possibility to directly identify the phosphorylated proteins or the kinases responsible of the phosphorylation. Nevertheless, this work allowed the authors to reveal the presence of Ser, Tyr and Thr phosphorylation in their Symbiodinium samples. They also revealed phosphorylation/dephosphorylation events in response to light stimuli. Despite the aforementioned limitations, this piece of work is interesting per se, especilly if we consider the very little amount of data already published in the field.

Additional comments

To my sense, this work meets Peer J standards and should be published after completion of the minor revisions listed herein.

Reviewer 2 ·

Basic reporting

The authors Castilla-Medina et al. have presented a profile of phosphorylated proteins from three species of the Symbiodinium family and specifically the light induced phosphorylation and rephosphorylation switch of a 75 kDa proteins from S. microadriaticum. The study is essential as it aimed at filling the knowledge gap about the key signal transduction proteins that undergo the phosphorylation and dephosphorylation switch in response to environmental stimuli like light conditions, nutrient content and availability, etc. The manuscript is easy to understand with clear English language and proper use of grammar and is structured as per PeerJ standards. The authors have reviewed relevant and adequate background information to set a context to the research question being addressed. The reviewer thanks the authors for providing the necessary raw data corroborating their results.

Experimental design

Though the authors have well-defined the research question and provided relevant data to corroborate their findings, the results are preliminary and require further corroboration with orthologous methods. The reviewer has following concerns about the data reported and suggestions to further improve the manuscript –
1. The authors have stated the detailed protocol for cell lysate preparation and isolation of supernatant for protein analysis, however, protease and phosphatase inhibitors were not used during lysate preparation. It is critical to use a concoction of these inhibitors, especially, phosphatase inhibitors as Espina, V. et al (Molecular & cellular proteomics, 7(10), 1998-2018) reported an augmentation of phosphorylation after phosphatase inhibitor use.
2. The study uses polyclonal anti-pThr and anti-pSer antibodies whereas monoclonal anti-pTyr. The authors should provide an explanation for the same.
3. The western blot procedure uses PBST, however, phosphates in the PBST buffer can potentially interfere and therefore it is recommended to use TBST instead of PBST
4. The authors should consider presenting the total protein content of the lysate as control.
5. The intensity of the bands indicated for phosphorylated proteins in figure 1 are very weak and barely visible. Perhaps, the authors can try lower dilutions of the primary and/or secondary antibodies, longer incubations times or temperature etc. to increase the intensity of the bands. However, if the authors have already tried different ways to improve the band intensity, they should mention in the results section.
6. To further support their western blot findings about the effect of light on dephosphorylation & rephosphorylation of pp75 protein in S. KB8, the authors should consider estimating the band intensities in figures 2 and 3 using densitometry analysis on three separate assays to estimate the statistical significance of this phenomenon.
7. Though the western blot is partially convincing, the authors should definitely consider using orthologous methods like mass spectrometry to corroborate their findings. Mass spectrometry is highly sensitive and is considered a gold standard method for studying phosphorylated proteins and other post-translational modifications (Mann, M. et al. (2002) Trends Biotechnol. 20:261).

Validity of the findings

The above-mentioned suggestions will strengthen the manuscript and will serve to fill a critical knowledge gap as mentioned by the authors. Though the findings are interesting, the work is still preliminary and needs further improvement before it can be accepted for publication. The reviewer strongly advocates use of orthologous method to corroborate the findings as, in general, there are some concerns with the specificity and cross-reactivity of phospho-antibodies.

Reviewer 3 ·

Basic reporting

The results presented were very limited and immature for publishing.

Please also see my comments below.

Experimental design

This manuscript falls within the scope of PeerJ, but the research question was vaguely defined. The hypothesis, approaches, and results were not justified.

The authors identified the gap as ‘…very little information from the genus Symbiodinium regarding proteins that participate in key phosphorylation processes for signal-transduction events that arise from environmental stimuli’. They sought to identify phosphorylated proteins, but showed only western blots, without actually identifying anything.

Why didn’t the author use the mass spec to identify the proteins? The author should at least report which genes/proteins are the phosphorylated proteins. Otherwise, this piece of research will not benefit anyone in the field or those who are interested in Symbiodinium biology.

The genome of Symbiodinium kawagutii was assembled and annotated in 2015 (Lin S. et al, Science). The authors should try to identify the protein of interest using this data. Are there homologous proteins to other known light-induced signaling proteins?

Validity of the findings

It is unclear how many times these western blots have been performed.

‘The intensity of band’ was not properly quantified and no meaningful replications was reported. Looking at fig2,3, S1, and S2, I can’t reach the same conclusion as the authors.

The reported changes in fig2 & fig3 are not validated. This change in phosphorylated protein level could result from a change in total protein level. If that is the case, then this process is not related to phosphorylation at all.

None of the results were validated by other methods. I am not convinced. The authors can’t even claim that they have been observing the same protein. Assume there are two proteins, both run at 75kD in a gel, one is upregulated and one is down-regulated, the method here won’t be able to distinguish them.

The conclusions are not well supported by the results.

---

## Round 0.2 · Minor Revisions

Please address the remaining issues raised by the reviewer and revise your manuscript accordingly.

Reviewer 2 ·

Basic reporting

The authors Castilla-Medina et al. have presented response to my previous reviews for their manuscript where they have investigated the phosphorylation/dephosphorylation response of Symbiodinium proteins to light stimuli. The revised manuscript is easy to understand with clear English language and proper use of grammar and is structured as per PeerJ standards. The authors have reviewed relevant and adequate background information to set a context to the research question being addressed. The reviewer thanks the authors for providing the necessary raw data for densitometric analysis.

Experimental design

The following is my response to their rebuttal –
1. We carried out all the experiments again now using protease and phosphatase inhibitors and all the results reported in this revised version are from experiments carried out again, now in the presence of protease and phosphatase inhibitors.
As suggested in my previous review, the authors performed all their experiments in presence of protease and phosphatase inhibitors. They detail the use of this inhibitor complex in the methods section. This reviewer thanks them for carrying out all the experiments again in the presence of protease and phosphatase inhibitors.
2. The use of the anti-phospho amino acid antibodies polyclonal or monoclonal depended exclusively on the commercial availability and advertised specificity. In this revised version, we provided the catalog numbers for each antibody we used.
I am contented with this explanation and thank the authors for providing the catalog numbers.

3. We found no difference in the immunodetection of the phosphoproteins between either buffer; the results were identical, and we provide the evidence as a new Figure S2.
The reviewer thanks the authors for providing this evidence that shows that there in no difference in using either of the buffers.

4. We used now as an internal loading control, the presence of actin; in this revised version we show the band intensities as well as the densitometric analysis normalized to this internal control (Fig. 3).
The use of actin as the internal loading control and the densitometric analysis have bolstered the authors claims.

5. The new experiments yielded much clearer bands and they are shown in the corresponding figures; additionally, labels were placed to identify all the detected bands (Figure S1).
This reviewer thanks for providing this new image with clearer bands, however, the authors should detail what was done differently to get clearer bands?

6. The re-phosphorylation effect was not observed in the new experiments with more rigorously temperature and light control, so all mention to this was deleted. On the other hand, the dephosphorylation effect was reproduced and densitometric analyses were carried out on triplicate samples from each species and the corresponding graphs are now shown as Figures 3 A-B. At least one of the time points in the three species analyzed showed a statistically significant decrease in phosphorylation after the light stimulus.
Thank you for providing the densitometric analyses and carrying out the experiments with more rigorous controls.

7. We carried out IEF-SDS-PAGE 2D analysis to isolate the corresponding spots (Figure 4 A-B and S5) and the partial peptide sequence, as well as the unequivocal identity of the protein is now reported (Figure 4C) This made also necessary a change in the title.
Thank you for including this new analysis which has significantly improved the quality of the manuscript.

Validity of the findings

The response and the experiments carried out by the authors adequately answer all the questions that were raised in the previous review. Expect for minor question I have in point 5, the manuscript has been significantly improved and meets the standards for publication in PeerJ. I recommend that this manuscript be accepted for publication after answering the minor question in point 5.

---

## Round 0.3 · accepted · Accept

All critiques are adequately addressed and therefore the revised manuscript can be accepted now.